# A longitudinal study of four-year changes in physical fitness among university students before and after COVID-19: 2019–2022

**Liangru Guo, Liang Jiang** *, **Huizhi Huang**

School of Sports Science, Hengyang Normal University, Hengyang, Hunan Province, China

* jiangliang_123@163.com

## Abstract

### Background

Physical fitness is a key indicator of health in youth and was significantly affected by the COVID-19 pandemic. However, longitudinal studies tracking pandemic-related fitness changes over four years in university students, particularly those enrolled before the pandemic, remain scarce. Understanding these long-term effects on body composition, fitness, and obesity prevalence—with attention to sex differences—is essential.

### Methods

This four-year cohort study included 4,413 Chinese university students (27% male, 73% female). From 2019 to 2022, participants underwent annual physical fitness assessments, measuring height, weight, vital capacity, long-jump, sit-reach, 50-m run, and sex-specific tests (1,000-m run/pull-ups for males; 800-m run/sit-ups for females). Statistical analyses included one-way ANOVA, LSD tests, and Chi-squared tests.

### Results

Weight and obesity rates increased significantly, with male obesity rising from 6.97% to 11.6% and female obesity from 1.9% to 3.45%. Overweight prevalence also grew slightly. Vital capacity peaked in 2021 but declined sharply in 2022 (males: 4,114.16 mL to 3,934.31 mL; females: 3,314.69 mL to 2,957.8 mL). Long-jump and sit-reach performance declined, and cardiorespiratory endurance (1,000-m/800-m runs) worsened post-2020. Female students improved in 50-m runs and sit-ups, while males showed no significant changes in 50-m runs or pull-ups.

**Data availability statement:** All relevant data are within the paper and its Supporting Information files.

**Funding:** (1)The Research Project on Teaching Reform of General Colleges and Universities in Hunan Province in 2023 (Project No. HNJG-20230872) Recipient: Liangru Guo (2) The Fourteenth Five-Year Plan of Educational Science in Hunan Province in 2024 (Project No. XJK24CTW013) Recipient: Liangru Guo (3)The Deepening Educational Evaluation Reform in the New Era in Hunan Province in 2022 Pilot Project (Project No. TGYX202207). Conceptualization, L.R.G. and L.J.; validation., L.R.G. and L.J.; formal analysis, L.R.G.; investigation, L.J.; resources, L.J.; writing—original draft preparation, L.R.G.; writing—review and editing, L.J.; supervision, L.J.; project administration, L.J.; funding acquisition, L.R.G and L.J. All authors have read and agreed to the published version of the manuscript.

**Competing interests:** The authors have declared that no competing interests exist.

## Conclusions

The pandemic significantly impacted university students' physical fitness over four years, exacerbating obesity risks and declining cardiorespiratory endurance, particularly in males. Schools should prioritize BMI management, obesity prevention, and targeted fitness interventions—emphasizing agility, strength, and late-stage cardiorespiratory training—to mitigate long-term health consequences.

## 1 Introduction

Coronavirus Disease 2019 (COVID-19) was first identified in 2019 and subsequently spread rapidly to most countries around the world in just a few months, becoming a global public health crisis [1]. The pandemic has not only put enormous pressure on global healthcare systems, but has also had a profound impact on human physical and mental health. An international online survey study covering multiple countries showed that home quarantine measures during the COVID-19 pandemic had a significant negative impact on all physical activity (PA) intensity levels, including vigorous, moderate, walking, and overall activity. In addition, it was found that people's daily sitting time increased significantly from an average of 5–8 hours during the pandemic, and that this increase in sedentary behavior severely impacted the public's level of physical fitness [2].

The university years represent a critical transition period from adolescence to adulthood, during which individuals establish long-term lifestyle and behavioral patterns [3]. However, the COVID-19 pandemic and its associated restrictions disrupted this formative phase. In China, strict containment measures—including prolonged campus closures and a rapid shift to online learning (December 2019–September 2020)—fundamentally altered daily life. While effective in curbing viral transmission, these policies inadvertently reduced PA levels among university students, a population particularly vulnerable to lifestyle-related health risks. Emerging evidence links sedentary behavior during the pandemic to adverse health outcomes, including obesity, hypertension, and metabolic disorders [4–7]. Among university students, studies report significant PA declines and weight gain due to restricted mobility and reduced sports participation [8]. Given PA's established role in preventing chronic diseases, these trends raise concerns about lasting health implications for this demographic.

Declining physical fitness has become a global social problem that has been further exacerbated, especially during the COVID-19 pandemic [9]. Several studies worldwide have focused on how the restrictions resulting from the COVID-19 pandemic exerted impacts on the physical fitness of children and adolescents, with the vast majority of findings suggesting significant negative impacts on physical fitness due to decreased PA and increased sedentary behavior [10–14]. However, recent studies have also revealed the complexity of this issue. One of these studies showed that not all health indicators were negatively affected by the lockdown, with body mass index (BMI), muscle strength, and flexibility qualities being the main ones affected [15]; another study pointed out that Chinese university students experienced

an increase in BMI both during and after the lockdown [16]. These two studies are important references: the first study conducted a one-year follow-up of the effects of the COVID-19 lockdown on BMI and physical fitness of Chinese university students; the second study analyzed the changes in BMI mainly over a three-year follow-up period.

Although these studies provide valuable insights, their shorter-term designs (1–3 years) limit the ability to distinguish between transient disruptions and sustained shifts in physical fitness. A four-year longitudinal study (2019–2022) offers unique advantages by capturing: (1) pre-pandemic baseline measures (2019), essential for isolating true pandemic effects; (2) acute-phase declines during strict lockdowns (2020); (3) post-lockdown recovery or adaptation patterns (2021–2022); and (4) potential sex- or policy-specific trajectories that shorter studies may overlook. Crucially, this extended timeframe allows us to assess whether observed changes represent temporary fluctuations or sustained trends—a distinction essential not only for public health planning but also for developing targeted strategies to prevent and treat long-term health outcomes related to PA.

This longitudinal study uniquely examines four-year changes in body composition (2019–2022) among university freshmen enrolled pre-pandemic (2019 cohort). We assessed body size through standardized anthropometric measurements (height, weight and BMI), alongside physical fitness tests. Our analysis focuses on temporal patterns and sex differences in overweight/obesity prevalence and fitness parameters, providing critical evidence to guide policymakers in preventing and mitigating long-term health outcomes associated with reduced PA under pandemic containment measures among student populations.

## 2 Materials and methods

The study was conducted in accordance with the Declaration of Helsinki, and approved by the Institutional Review Board of the School of Sports Science, Hengyang Normal University OF INSTITUTE(September 18, 2019).

### 2.1 Study population

The recruitment period for this study spanned from September 5, 2019, to September 28, 2019. A total of 4,413 subjects, including 1,206 male and 3,207 female students (mean age = 17.90 years, SD = 0.90), were selected from freshmen enrolled during this period at a teacher training university in Hunan Province and were followed up for four years throughout their university study.

The study utilized a convenience sample consisting of incoming freshmen from the class of 2019 who were both available and willing to participate. Inclusion required students to provide written informed consent and complete baseline health and demographic questionnaires. Individuals with pre-existing chronic health conditions—such as cardiovascular disease or severe musculoskeletal disorders—or other physical limitations affecting exercise participation were excluded. Despite this, the sample maintained representativeness by encompassing all consenting students from the admitted cohort, thereby reducing potential selection bias.

All physical fitness tests were conducted using uniform, calibrated measuring devices to ensure the comparability of results across different testing periods. According to the "China Student Physical Fitness and Health Research Workbook" [17], the physical fitness and health data of these university students were investigated in 2019 (freshman stage, before the pandemic), 2020 (sophomore stage, during the pandemic), 2021 (junior stage, after the pandemic), and 2022 (senior stage, after the pandemic). A total of 4572 students participated in the baseline survey in 2019. Over the four-year period, 159 participants were lost to follow-up due to [briefly state reasons, e.g., voluntary withdrawal, transfer, or incomplete testing], resulting in a final analytical sample of 4413 students (1206 males and 3207 females) who completed all assessments across all four years, representing a retention rate of 96.5%. Informed consent was obtained from all participants prior to inclusion in the study.

### 2.2 Physical fitness test

To ensure consistency in data collection, all physical fitness tests throughout the four-year study period were administered by the same team of trained physical education professors. Prior to the study, all test administrators underwent a

standardized training protocol to ensure uniform assessment procedures. Physical fitness tests for both male and female included height, weight, vital capacity, long-jump, sit-reach, and 50-m run. Sex-specific tests were included to account for physiological differences: pull-ups and a 1,000-m run for male assess upper-body strength and endurance, while sit-ups and an 800-m run for female measure core strength and aerobic capacity. To ensure measurement consistency, all subjective assessments were conducted following unified training, with standardized scoring criteria strictly enforced among all testers. Physical fitness tests were administered by physical education professors and conducted annually from October through November. All tests were performed under consistent conditions, including indoor venues with standardized equipment (temperature maintained at 20–24°C and relative humidity at 40–60%), and scheduled at the same time of day (morning sessions) to minimize environmental variability. Potential variations were documented and accounted for in the statistical analysis. Before each fitness test, participants completed a standardized 15-minute warm-up routine consisting of light jogging (5 minutes), dynamic stretching (5 minutes for major muscle groups), and sport-specific activation exercises (5 minutes). To maintain adherence to protocols, test administrators received annual training, and participant engagement was monitored through attendance records and real-time supervision during testing.

**2.2.1 Vital capacity.** The vital capacity was measured using an air spirometer (GMCS-IV; Jianmin, Beijing, China) with a dry and sterilized plastic mouthpiece. Participants took a deep breath and slowly exhaled into the mouthpiece until they could no longer exhale. The air spirometer automatically calculated the maximum air volume (vital capacity per milliliter) and displayed the result upon completion of the exhalation maneuver. Participants performed the test twice with a 15-second interval between trials, and the higher of the two values, as determined by the standardized scoring protocol, was recorded.

**2.2.2 Long-jump.** Participants were asked to stand behind the starting line with their feet apart and jump forward in a natural position. Participants were also instructed that both feet should jump at the same time from a standing position and that no additional movements should be made. Horizontal distance was measured from the trailing edge of the starting point to the trailing edge of the nearest landing point. Each student was allowed for three attempts. The longest jump was recorded in centimeters.

**2.2.3 Sit-reach.** Participants were tested for sitting and stretching using a seated stretch box device, in a seated position and keeping their legs straight. The participant's upper body was leaned forward, with his/her arms straightened forward and his/her feet separated by 10–15 centimeters while pushing horizontally forward against the test board (GMCS-IV; Jianmin, Beijing, China). The participant's fingertips extended and gradually pushed the test bar forward until he/she was unable to push further. Each student was given two attempts. The farthest distance was recorded in centimeters.

**2.2.4 Pull-ups.** The test is aimed to check a participant's upper body strength. The participant held the bar in a shoulder-width hand to form a straight-arm suspension position, and then pulled the entire weight upward simultaneously using only the arms while keeping the body and arms straight. In addition, the participant's body could not make additional movements, such as swinging or oscillating. In order to complete a pull-up, the participants were required to pull themselves up until their chins cleared the horizontal level of the bar. The final number of correctly completed pull-ups was recorded. No strict time limit was imposed; participants performed pull-ups continuously until exhaustion.

**2.2.5 Running tests (50-m, 800-m, and 1,000-m Runs).** 50-m Run: Participants began in a standing position at the starting line of a 50-meter straight track. Upon the physical education professor's verbal signal ("ready-and-go"), the timekeeper started recording. Time stopped when the participant's chest crossed the finish line. The recorded time (in seconds, to one decimal place) was rounded up if the second decimal digit was non-zero. Each participant completed two trials, with the better performance was recorded.

800-m Run: Female students performed an 800-meter run, starting behind the line upon the physical education professor's "ready-and-go" command. Time (in minutes:seconds, to one decimal place) was recorded upon chest crossing the finish line, with the recorded time rounded to the nearest tenth of a second. Only one attempt was allowed.

1,000-m Run: Male students completed a 1,000-meter run under identical protocols to the 800-m and 50-m runs regarding starting procedure, timing method, and rounding rule (single attempt, recorded time rounded to the nearest tenth of a second).

**2.2.7 Body mass index (BMI).** BMI was used to measure the degree of overweight and obesity of the participants, expressed as weight (kg)/height (m)$^2$. The WHO-defined BMI criteria were used, with a BMI ≥ 28 (kg/m$^2$) defined as obesity and 24 ≤ BMI < 27.9 defined as overweight. Participants' height and weight were measured by using electronic weight and height scales. Participants were instructed to remove their shoes and wear light indoor clothing before standing upright on the electronic scale. The scale then automatically measured weight and height to the nearest 0.1 cm.

## 2.3 Statistical analysis

Data were initially entered using Excel 2016 and statistically analyzed using the software SPSS 27.0; quantitative data were expressed using mean ± standard deviation. Given the repeated measurements across multiple years, one-way analysis of variance (ANOVA) was selected to compare adjacent years stratified by sex, as it provides a straightforward method for detecting interannual differences while accounting for within-group variability. Potential confounding variables such as age, baseline fitness, and socioeconomic status were not explicitly adjusted for in the analysis. The $\chi^2$ test in SPSS 27.0 was used to analyze differences in the detection rates of overweight and obesity in the same year for different sex groups; and $\chi^2$ linear trend test was used to analyze the trends in the detection rates of overweight and obesity in the whole population, both sex, for the years 2019–2022. The annual growth rate (%/year) of the detection rate of each sex-age group in different periods was calculated (or folded): Annual growth rate = (detection rate in a certain year – detection rate in the previous year)/ (a certain year – the previous year) × 100%. To ensure measurement consistency, all physical fitness testing procedures were standardized, and trained personnel conducted the assessments using calibrated equipment. A two-sided test was used, and a difference of P < 0.05 was considered statistically significant. In this study, data distribution was visualized using box plots (also known as box-and-whisker plots), which display the minimum, lower quartile (Q1), median (Q2), upper quartile (Q3), and maximum values. The box plots were generated using Microsoft Excel.

## 3 Results

### 3.1 Demographic characteristics of participants

Table 1 summarizes the demographic characteristics of the participants: female students outnumbered male students (72.7% to 27.3%); their age was calculated based on the enrollment in September 2019, with the 17 ≤ YEAR ≤ 19 year old age group accounting for 91.9% of the total participants, followed by the < 17 year old age group accounting for 4.7% and the > 19 year old age group accounting for 3.4%; 89.2% of the university students were Han Chinese and 10.8% were ethnic minorities.

Table 1. Demographic characteristics.

| Variable | Category | Frequency | Percent valid | Effective percentage | Cumulative percentage |
|---|---|---|---|---|---|
| Sex | Male | 1,206 | 27.3 | 27.3 | 27.3 |
| | Female | 3,207 | 72.7 | 72.7 | 100.0 |
| Age | <17 | 208 | 4.7 | 4.7 | 4.7 |
| | 17 ≤ year ≤ 19 | 4,055 | 91.9 | 91.9 | 96.6 |
| | >19 | 150 | 3.4 | 3.4 | 100.0 |
| Ethnicity | Han Chinese | 3,935 | 89.2 | 89.2 | 89.2 |
| | Ethnic minority | 478 | 10.8 | 10.8 | 100.0 |

### 3.2. Results of ANOVA for university students' physical fitness

**3.2.1 Changes in body shape.** In terms of height changes, the male group did not show a significant difference in the mean height between the four years ($F = 1.745$, $p > 0.05$). In contrast, the female group showed a significant difference in the mean height ($F = 7.375$, $p < 0.01$). The LSD test, a post-hoc test, showed that the difference in the height of the females was not significant between 2019 and 2020, nor between 2021 and 2022, but those in the latter two years (2021–2022) were significantly higher than those in the first two years (2019–2020) (Table 2).

Regarding weight changes, there was a significant difference in the mean weight of the male group ($F = 19.698$, $p < 0.01$). LSD analysis showed that there was no significant difference in weight between 2019 and 2020, but there was a trend of increasing year by year from 2021 to 2022. A significant difference was also observed in females ($F = 11.446$, $p < 0.01$). Their weight remained stable between 2019 and 2021 but increased significantly in 2022 compared to the preceding three years (Table 2).

In terms of BMI, there was a significant difference in the males ($F = 20.095$, $p < 0.01$), whose BMI remained stable between 2019–2021 and was significantly higher in 2022. A similar trend was observed in the females ($F = 8.269$, $p < 0.01$), whose BMI did not change significantly between 2019, 2020 and 2021 and was significantly higher in 2022 than those in the previous three years. (Table 2)

**3.2.2 Changes in physical fitness.** Significant differences were observed across years in the vital capacity test for both males ($F = 132.75$, $p < 0.01$) and females ($F = 941.83$, $p < 0.01$). In males, vital capacity increased from 2019 to 2021, peaked in 2021, and then declined in 2022. Females showed lower values during 2019–2020 compared to 2022, with both periods being significantly lower than the peak in 2021.

In the 50-m run test, no significant difference was found in males ($F = 0.69$, $p > 0.05$), whereas a significant difference was observed in females ($F = 1,187.2$, $p < 0.01$), with performance remaining stable during 2019–2021 and improving significantly in 2022.

Both the long-jump and sit-reach tests revealed similar trends across sexes. In the long-jump, significant differences were detected in males ($F = 128.238$, $p < 0.01$) and females ($F = 988.04$, $p < 0.01$). Male scores were lower in 2020 than in 2022, and both were lower than in 2019 and 2021, with no difference between the latter two years. Female scores were significantly lower in 2019 and 2022 than in 2020, and all three years were lower than 2021, with no difference between 2019 and 2022. For the sit-reach test, both males ($F = 11.032$, $p < 0.01$) and females ($F = 144.36$, $p < 0.01$) exhibited a decreasing trend over time, with no significant difference between 2019 and 2020 in either group.

Similarly, the endurance running test (1,000-m for males, $F = 246.005$, $p < 0.01$; 800-m for females, $F = 11.161$, $p < 0.01$) showed comparable patterns: performance was significantly better in 2019 and 2020 than in 2021 and 2022, with no significant differences within the earlier or later pairs of years.

Additionally, a significant difference was found in the 1-min sit-ups test for females ($F = 912.35$, $p < 0.01$), with scores in 2019 and 2020 being lower than in 2021, and all three years lower than 2022, and no difference between 2019 and 2020. (Table 2)

### 3.3 Detection of overweight and obesity among university students

**3.3.1 Detection of obesity.** In the longitudinal study on the university students' obesity detection rates, an increasing trend year by year was observed. The obesity detection rate for the males increased from 6.97% in 2019 to 11.6% in 2022, showing a linear increase over time [$\chi^2$ (trend)=18.44, $P < 0.001$]. The obesity detection rate for the females increased from 1.9% in 2019 to 3.45% in 2022, with a linear increase over time [$\chi^2$ (trend)=19.81, $P < 0.001$]. The figure for the whole population increased from 3.29% to 5.69%, indicating a linear trend over time [$\chi^2$ (trend)=37.39, $P < 0.001$]. The boxplot analysis revealed a progressive increase in BMI values across the study period (2019–2022) for both sex, with a clear upward trend observed year by year (Figs 1 and 2). Sex-stratified $\chi^2$ tests revealed significantly higher obesity

**Table 2. Results of ANOVA.**

| Variable | Sex | N | 2019 | 2020 | 2021 | 2022 | F | LSD |
|---|---|---|---|---|---|---|---|---|
| Height | Male | 1206 | 172.50 (6.15) | 172.50 (6.15) | 172.93 (6.23) | 172.89 (6.20) | 1.75 | |
| | Female | 3207 | 160.15 (5.48) | 160.15 (5.48) | 160.56 (5.62) | 160.64 (5.51) | 7.36** | 1&2<3&4 |
| Weight | Male | 1206 | 65.75 (12.33) | 65.75 (12.33) | 66.91 (12.68) | 69.16 (12.95) | 19.70** | 1&2<3<4 |
| | Female | 3207 | 53.96 (7.54) | 53.92 (7.74) | 53.96 (7.94) | 54.90 (8.73) | 11.45** | 1&2&3<4 |
| BMI | Male | 1206 | 22.06 (3.75) | 21.07 (2.67) | 20.94 (2.79) | 21.22 (3.01) | 20.10** | 1&2&3<4 |
| | Female | 3207 | 21.38 (3.17) | 21.00 (2.68) | 21.97 (3.57) | 21.26 (3.10) | 8.27** | 1&2&3<4 |
| Vital capacity | Male | 1206 | 3880.37 (475.95) | 3778.78 (459.56) | 4114.16 (446.12) | 3934.31 (283.98) | 132.75** | 2<1<4<3 |
| | Female | 3207 | 2939.28 (337.19) | 2941.09 (336.51) | 3314.69 (343.61) | 2957.80 (344.47) | 941.84** | 1&2<4<3 |
| 50-m run | Male | 1206 | 7.16 (0.48) | 7.16 (0.48) | 7.18 (0.46) | 7.18 (0.47) | 0.69 | |
| | Female | 3207 | 9.39 (0.42) | 9.39 (0.42) | 9.38 (0.42) | 8.91 (0.31) | 1,187.22** | 4<1&2&3 |
| Long-jump | Male | 1206 | 228.94 (14.68) | 218.18 (18.54) | 229.19 (18.94) | 221.75 (14.17) | 128.24** | 2<4<3&1 |
| | Female | 3207 | 161.80 (13.33) | 163.06 (12.59) | 176.55 (13.70) | 161.21 (13.00) | 988.04** | 1&4<2<3 |
| Sit-reach | Male | 1206 | 16.07 (6.44) | 16.07 (6.44) | 15.49 (7.34) | 14.7 (6.85) | 11.03** | 4<3<1&2 |
| | Female | 3207 | 19.63 (6.00) | 19.63 (6.00) | 18.5 (6.30) | 16.89 (6.08) | 144.36** | 4<3<1&2 |
| 1,000-m-run | Male | 1206 | 236.83 (23.54) | 236.83 (23.54) | 253.19 (17.57) | 252.92 (17.46) | 246.01** | 1&2<3&4 |
| 800-m-run | Female | 3207 | 251.69 (17.85) | 251.69 (17.85) | 253.6 (17.10) | 253.36 (17.35) | 11.16** | 1&2<3&4 |
| Pull-ups | Male | 1206 | 12.16 (3.53) | 12.16 (3.53) | 12.23 (3.48) | 12.24 (3.35) | 0.17 | |
| 1-min-sit-ups | Female | 3207 | 35.06 (5.66) | 35.21 (5.60) | 39.54 (5.78) | 41.1 (5.88) | 912.35** | 1&2<3<4 |

Note: **P<0.01; 1=2019; 2=2020; 3=2021; 4=2022. Body mass index (BMI)

rates in males versus females annually (P<0.01). Notably, 2020 (first COVID-19 year) showed no significant obesity rate increase for either sex (Table 3).

Annual obesity rate growth rates demonstrated sex-specific patterns: males (−1.00%, 2.49%, 3.14%) and females (−0.03%, 0.78%, 0.81%) between adjacent years. Both sexes showed slight declines in 2019–2020 followed by steady increases in 2020–2022, with males exhibiting consistently higher growth rates.

**3.3.2 Detection of overweight.** In the longitudinal study on the university students' overweight detection rates, the overweight detection rate of the male university students increased from 19.98% in 2019 to 20.9% in 2022, and that of the female university students increased from 10.98% in 2019 to 11.85% in 2022. The figure for the entire population increased from 13.43% to 14.32%. In terms of sex stratification, the results of $\chi^2$ test showed that the difference in overweight detection

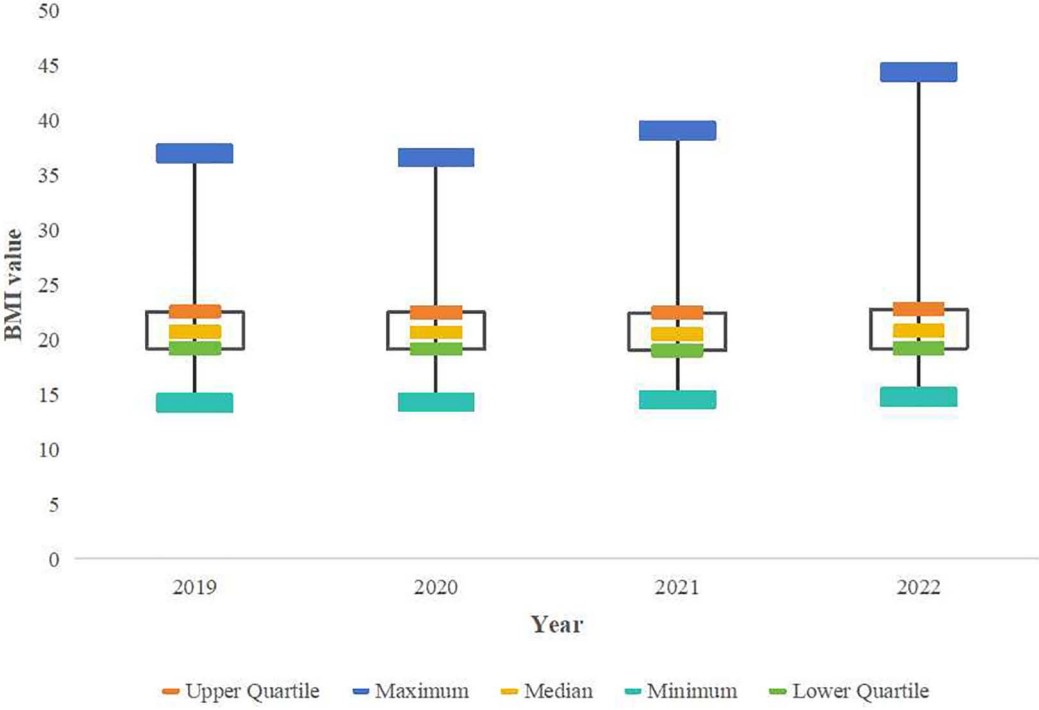

**Fig 1. Boxplot of Female BMI Distribution (2019–2022).**

rates between male and female university students each year was statistically significant (P<0.01), with the males having a higher overweight detection rate than the females. It is worth noting that there was no significant increase in the overweight detection rate in the male and female groups from 2020, which was the first year of the COVID-19 pandemic. The overweight detection rate showed a significant decrease in 2021 and a significant increase in 2022 (Table 4).

Between 2019 and 2022, the annual growth rates of the overweight detection rate among the males were −0.18%, −1.89%, and 2.99% between adjacent years, respectively, and the figures for the females were −0.57%, −1.09%, and 2.53% between adjacent years, respectively. The annual growth rates of the overweight detection rate of both the males and the females showed a decreasing trend from 2019 to 2021, and continued to increase from 2021 to 2022, with the annual growth rate of the males being much higher than that of the females.

## 4 Discussions

This longitudinal study provides the first evidence of COVID-19's impact on physical fitness trends among Chinese university students over four years (2019–2022). Key findings revealed sex-specific changes: females showed significant temporal differences in height, weight, and BMI, while males exhibited changes only in weight and BMI. Physical function (e.g., vital capacity) and fitness metrics (e.g., long jump, sit-reach) declined significantly in both sexes, with females also showing reductions in 50-m run and sit-up performance. Notably, obesity rates increased annually, particularly during the 2020 pandemic peak, with males consistently demonstrating higher overweight/obesity rates than females. Overweight rates initially declined (2019–2021) but rebounded in 2022.

### 4.1 Changes in the body size of college students

The results on changes in the body size have similarities with the findings of a number of international studies that have documented the negative impact of the COVID-19 pandemic on the body size. Numerous studies similarly found an

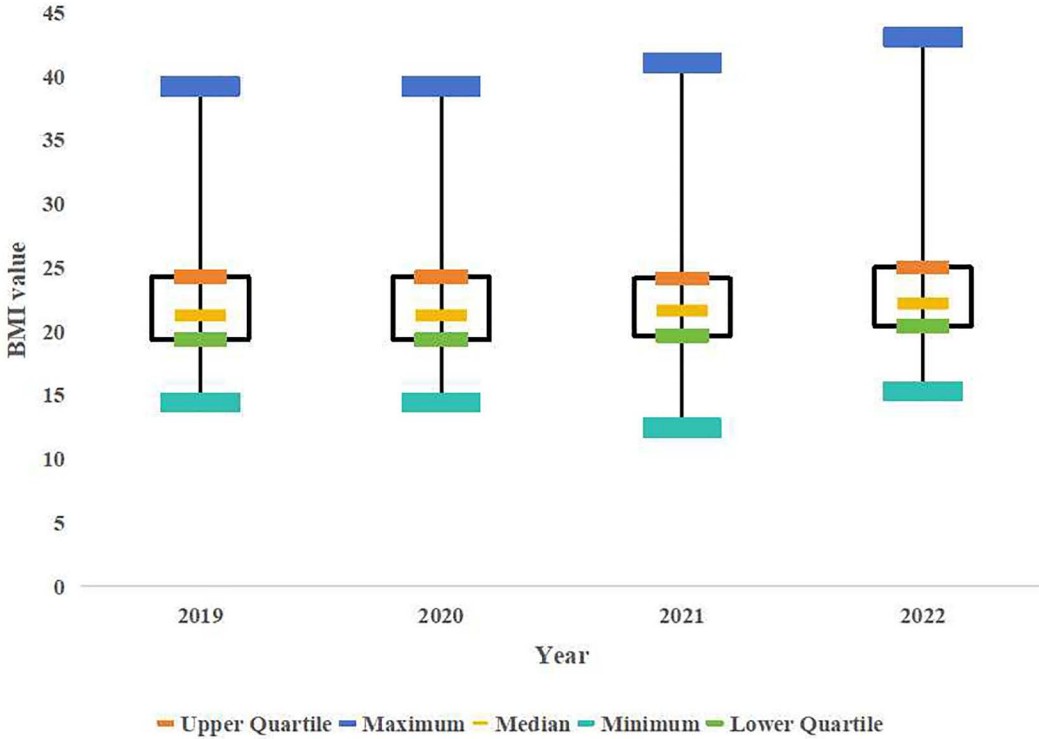

**Fig 2. Boxplot of Female BMI Distribution (2019–2022).**

**Table 3. Comparison of obesity detection rates.**

| Groups | 2019 | | | 2020 | | | 2021 | | | 2022 | | | χ²(trend) |
|---|---|---|---|---|---|---|---|---|---|---|---|---|---|
| | Sta-tis-tical value | Total num-ber of people | Number of detections (detection rate) | Statis-tical value | Total num-ber of people | Number of detections (detection rate) | Statis-tical value | Total num-ber of people | Number of detections (detection rate) | Sta-tis-tical value | Total num-ber of people | Number of detections (detection rate) | |
| Male | | 1206 | 84 (6.97%) | | 1206 | 84 (5.97%) | | 1206 | 102 (8.46%) | | 1206 | 140 (11.6%) | 18.440** |
| Female | | 3207 | 61 (1.90%) | | 3207 | 60 (1.87%) | | 3207 | 85 (2.65%) | | 3207 | 111 (3.46%) | 19.810** |
| | χ² | 70.7** | | χ² | 72.05** | | χ² | 72.84** | | χ² | 108.46** | | |
| the total | | 4413 | 145 (3.29%) | | 4413 | 144 (3.26%) | | 4413 | 187 (4.24%) | | 4413 | 251 (5.69%) | 37.39** |

Note: **P<0.01.

increase in BMI and a decrease in physical fitness among university students following the COVID-19 lockdown due to prolonged social distancing and school closures [18,19]. In our analysis, these disruptions—including lockdowns limiting outdoor activity and online learning reducing structured physical education—were explicitly considered as key contextual factors explaining the 2020 obesity surge. Although the COVID-19 pandemic appeared to be receding in 2022, the obesity and metabolic dysregulation pandemic continues to rage [20]. These trends are consistent with the fact that weight, BMI, and obesity rates among the studied population of university students showed a significant yearly increasing trend over time. In addition, the present study also found that, between sexes, the detection rates of overweight and obesity were

Table 4. Comparison of overweight detection rates.

| Groups | 2019 | | | 2020 | | | 2021 | | | 2022 | | | $\chi^2$(trend) |
|---|---|---|---|---|---|---|---|---|---|---|---|---|---|
| | Statistical value | Total number of people | Number of detections (detection rate) | Statistical value | Total number of people | Number of detections (detection rate) | Statistical value | Total number of people | Number of detections (detection rate) | Statistical value | Total number of people | Number of detections (detection rate) | |
| Male | | 1206 | 241 (19.98%) | | 1206 | 241 (19.8%) | | 1206 | 216 (17.91%) | | 1206 | 252 (20.9%) | 0.897 |
| Female | | 3207 | 352 (10.98%) | | 3207 | 334 (10.41%) | | 3207 | 299 (9.32%) | | 3207 | 380 (11.85%) | 0.537 |
| | $\chi^2$ | 61.131** | | | $\chi^2$ | 120.793** | $\chi^2$ | 62.694** | | $\chi^2$ | 58.454** | | |
| the total | | 4413 | 593 (13.43%) | | 4413 | 575 (13.03%) | | 4413 | 515 (11.67%) | | 4413 | 632 (14.32%) | 0.57 |

Note: **$P < 0.01$

much greater among the male students than among the female students during the four years of university, a finding that is also consistent with the findings of most previous studies [16,21,22]. This phenomenon may be related to the fact that females are more body-conscious and therefore less likely to gain weight during the pandemic than males [16].

However, it is worth noting that a previous study noted a significant increase in adolescent obesity detection rates in 2020 compared to 2019, with a decreasing trend in 2021 and 2022 [23]. The reason for this discrepancy may be related to the age range of the sample size. The respondents of this study were university students with an average age of 18–20 years. The lack of teacher supervision and insufficient self-discipline may have led to an increase in the obesity detection rate year after year. Previous studies have focused on the adolescent population aged 6–14 years, and their conclusions may differ from those from the university students. In addition, the results of this study showed that the overweight detection rate among the university students showed a decreasing trend from 2019 to 2021 and was significantly higher in 2022 than in the previous three years. This finding differs from the results of a previous study that found an increasing trend in overweight detection rates among university students from 2019 to 2021 [24]. This difference may be related to sample selection. The previous study focused on data from 10 universities in Hunan Province, whereas this study selected only one teacher training university in Hunan Province, where the female to male ratio was approximately 3:1. This substantial difference in the gender composition of the sample may be one of the reasons for the inconsistent results. Compared to pre-pandemic university populations without COVID-19 influence, our cohort exhibited intensified overweight/obesity trends, particularly in males. This acceleration suggests pandemic-specific impacts (e.g., prolonged campus lockdowns, reduced PA opportunities) exacerbated existing risk factors. However, the female-majority sample may have moderated overall trends, as females generally maintained lower obesity rates despite similar environmental pressures.

### 4.2 Changes in the physical fitness of university students

Regarding the results of changes in physical fitness, this study fully discussed from three perspectives: cardiorespiratory fitness, muscular strength and flexibility. From the perspective of cardiorespiratory fitness, the scores of both sex from the 1,000-m run (male) and 800-m run (female) were significantly better in 2019 and 2020 than in 2021 and 2022. This change may be related to the characteristics of China's education system. In China's higher education system, freshmen and sophomores are required to complete a certain number of physical education credits, whereas juniors and seniors are no longer subjected to a mandatory requirement for physical education [25]. In addition, PA and physical fitness (PF) levels of university students decline with age, with PF increasing from first to second year but gradually slowing from third to fourth year [26]. This finding is consistent with the results of a previous study, which noted a decreasing trend in endurance quality among university students one year after the implementation of COVID-19 restrictions (2020) [27,28]. From

the perspective of muscular strength, the results of this study showed that the male students' performance in the 50-m run and pull-ups did not change significantly over the four-year period, suggesting that male students' explosive and upper body strength remained stable during the COVID-19 pandemic. In contrast, the female students' performance in the 50-m run was significantly better in 2022 than in the previous three years, and their performance in 1-min sit-ups showed a year-to-year increase. These sex-specific improvements, particularly among females, may be partially attributed to social norms and increased familiarity with structured exercise in a university setting. As previous research in both civilian and military contexts has suggested, women often exhibit substantial gains in physical performance when provided with equal opportunities for training, potentially due to a lower initial baseline and thus greater room for improvement [29]. In addition, long-jump scores were significantly lower in freshmen and seniors than in sophomores and juniors. These results suggest that muscle strength did not significantly decrease in both the males and the females as a result of COVID-19. This phenomenon may be attributed to the fact that during home quarantine at the beginning of the pandemic, the university's physical education program required students to perform daily resistance training, including coursework such as push-ups, sit-ups, and weighted squats. Although the course was conducted online, it was still supervised online by a teacher. In addition, in China's higher education system, coursework accounts for 20% of the score for physical education classes and counts toward a student's grade point average (GPA) [25]. This system may have motivated students to maintain a certain amount of exercise during the pandemic, thus maintaining muscle strength levels. From the perspective of flexibility, this study found a decreasing trend in flexibility indicators over time for both the male and female students. This result is consistent with previous studies suggesting that home quarantine during a pandemic led to a decrease in flexibility among university students [15,30]. For example, a study identified the lack of systematic stretching training during the pandemic as a major cause of decreased flexibility [31]. In addition, the limited range of daily activities during home quarantine may have led to decreased joint mobility and muscle stretching, especially the iliopsoas muscle tightness of students due to prolonged sedentary periods resulting from online courses [32]. These changes highlight the long-term risk of reduced PA during pandemic disruptions, as flexibility declines may persist even after lockdowns end, potentially increasing injury risk and impairing overall mobility.

### 4.3 Practical implications

The findings of this study have important practical implications for public health and health promotion. The findings emphasize the importance of maintaining a healthy lifestyle during a pandemic, especially in terms of diet and PA. Second, the findings suggest the need for educational institutions and public health authorities to develop targeted interventions to address the growing of obesity in the university students. Finally, this study provides a scientific basis for future health policy development, especially on how to balance pandemic prevention and control with health promotion.

### 4.4 Limitations and future research directions

This study is subject to some limitations, although it study provides valuable data on the impact of the COVID-19 pandemic on the physical fitness of university students. First, the samples of this study were university students in a certain region of China, which may not fully represent the overall situation of university students nationwide. Second,while our longitudinal design spanned four years, missing data due to student attrition or incomplete testing records may affect result robustness. Additionally, the study data mainly relied on school physical examination data, which may have some measurement errors. Moreover, potential confounding variables such as age, baseline fitness, and socioeconomic status were not explicitly adjusted for in the analysis, which could potentially influence the outcomes and their interpretation. Furthermore, due to the large number of test items and participants, multiple data collectors were involved throughout the study period. Despite efforts to standardize protocols, this may have introduced inter-rater variability. Finally, the significant gender disproportion in the study population (with females being overrepresented) may have introduced bias and skewed the results, necessitating caution in the interpretation of the findings.

Based on the limitations of this study, future research can be conducted in the following ways: first, expand the sample size to cover more regions and different types of colleges and universities, so as to improve the representativeness and generalizability of the study. Second, more accurate measurement tools, such as accelerometers and body composition analyzers, can be used to reduce measurement errors. Finally, future studies could further explore the relationship between mental health and physical fitness during the pandemic, especially the effects of psychological factors such as anxiety and depression on body weight and physical fitness.

## 5 Conclusions

This study finds that COVID-19 significantly impacted the physical fitness and health of 2019-enrolled university students: obesity rates (especially among males) rose annually; flexibility declined consistently; endurance remained stable in the first two years but dropped in junior and senior years; and strength was unaffected. We recommend targeted interventions to control BMI (with focus on males), enhance agility, explosiveness, and muscular strength across all stages, and prioritize cardiorespiratory endurance improvement in the third and fourth years. These findings offer insights for exploring school closure/online learning effects on youth fitness and support policy-making for adolescent health. Limitations include a single cohort focus, and future research could expand to multi-cohort or longitudinal analyses to generalize conclusions.

## Supporting information

**S1 File. Supplementary material.**
(DOCX)

## Author contributions

**Conceptualization:** Liangru Guo, Liang Jiang.

**Data curation:** Liangru Guo.

**Formal analysis:** Liangru Guo.

**Funding acquisition:** Liang Jiang.

**Investigation:** Liang Jiang, Huizhi Huang.

**Methodology:** Huizhi Huang.

**Project administration:** Liangru Guo.

**Software:** Huizhi Huang.

**Supervision:** Liang Jiang.

**Writing – original draft:** Liangru Guo.

**Writing – review & editing:** Liang Jiang.

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
