## [Decision Letter · Decision Letter 0]

26 Jul 2025

PONE-D-25-29142A longitudinal study of four-year changes in physical fitness among university students before and after COVID-19: 2019-2022PLOS ONE

Dear Dr.  Jiang,

Thank you for submitting your manuscript to PLOS ONE. After careful consideration, we feel that it has merit but does not fully meet PLOS ONE’s publication criteria as it currently stands. Therefore, we invite you to submit a revised version of the manuscript that addresses the points raised during the review process.

We look forward to receiving your revised manuscript.

Kind regards,

Mohamed Ahmed Said, Ph.D.

Academic Editor

PLOS ONE

Journal Requirements:

3.Thank you for stating the following financial disclosure:

“This research was funded by the Research Project on Teaching Reform of General Colleges and Universities in Hunan Province in 2023 (Project No. HNJG-20230872), the Fourteenth Five-Year Plan of Educational Science in Hunan Province in 2024 (Project No. XJK24CTW013), and the Deepening Educational Evaluation Reform in the New Era in Hunan Province in 2022 Pilot Project (Project No. TGYX202207).”

4. Please note that funding information should not appear in any section or other areas of your manuscript. We will only publish funding information present in the Funding Statement section of the online submission form. Please remove any funding-related text from the manuscript.

7. Please remove all personal information, ensure that the data shared are in accordance with participant consent, and re-upload a fully anonymized data set.

Additional guidance on preparing raw data for publication can be found in our Data Policy (https://journals.plos.org/plosone/s/data-availability#loc-human-research-participant-data-and-other-sensitive-data) and in the following article: http://www.bmj.com/content/340/bmj.c181.long .

9.If the reviewer comments include a recommendation to cite specific previously published works, please review and evaluate these publications to determine whether they are relevant and should be cited. There is no requirement to cite these works unless the editor has indicated otherwise. 

Reviewers' comments:

Reviewer's Responses to Questions

**Comments to the Author**

1. Is the manuscript technically sound, and do the data support the conclusions?

Reviewer #1: Yes

Reviewer #2: Yes

Reviewer #3: Partly

Reviewer #4: Partly

Reviewer #5: No

2. Has the statistical analysis been performed appropriately and rigorously? 

Reviewer #1: Yes

Reviewer #2: Yes

Reviewer #3: Yes

Reviewer #4: I Don't Know

Reviewer #5: No

3. Have the authors made all data underlying the findings in their manuscript fully available?

Reviewer #1: Yes

Reviewer #2: Yes

Reviewer #3: No

Reviewer #4: Yes

Reviewer #5: No

4. Is the manuscript presented in an intelligible fashion and written in standard English?

Reviewer #1: Yes

Reviewer #2: Yes

Reviewer #3: No

Reviewer #4: Yes

Reviewer #5: No

5. Review Comments to the Author

Reviewer #1: General Assessment:

The manuscript presents original research that investigates changes in body composition and physical fitness among Chinese university students over a four-year period spanning before, during, and after the COVID-19 pandemic. The topic is of public health relevance, and the authors have collected a large dataset with multiple fitness indicators. Overall, the study meets the journal’s criteria for technical rigor and ethical compliance. However, several aspects require minor revisions to improve the clarity, consistency, and presentation of the work.

1. Scientific Rigor and Methods:

The cohort design is appropriate for the research objectives, and the repeated physical fitness testing across four years is a strength.

The statistical approach is acceptable for basic trend analysis. Nevertheless, the rationale for selecting ANOVA over more robust longitudinal models could be briefly explained. Additionally, more detail on data collection quality control (e.g., measurement consistency) is encouraged.

2. Tables and Figures:

Tables 2, 3, and 4 require formatting adjustments for clarity. Some rows and columns are misaligned, and headers are difficult to follow. Similarly, Figures 1 and 2 do not effectively communicate the temporal changes in BMI. Consider replacing the 3D scatterplots with simpler and more interpretable line charts or boxplots.

3. Discussion Section:

The manuscript lacks a focused discussion section. Beyond reporting results, the paper should include critical interpretation, explanations for observed trends, comparison with existing studies, and acknowledgment of limitations. A structured discussion would greatly improve the manuscript’s clarity and depth.

4. Writing and Organization:

The manuscript is generally well written in standard English. Minor language polishing could improve clarity and readability. The structure would benefit from clearer section transitions and concise topic sentences in the results and discussion.

5. Ethics and Data Availability:

Ethical approval and informed consent are clearly stated.

Data availability is appropriately addressed and aligns with PLOS ONE standards.

Recommendation:

The study is scientifically sound and methodologically appropriate. With minor revisions focused on improving table formatting, figure clarity, and manuscript organization, it will be suitable for publication.

Reviewer #2: Dear Authors,

Congratulations on your choice of topic. I read the article with interest. The manuscript provides important information on the physical fitness of students in the context of the COVID-19 pandemic. The study is based on measurements of physical fitness (tests), body weight and height, and spirometry. The description lacks information on whether the researchers conducting the study had been trained beforehand. The paper also does not indicate whether the study was conducted using uniform, calibrated measuring devices, which seems important from the point of view of the comparability of the results. The study has its limitations, but the authors have drawn attention to them by describing them in the limitations section. In their recommendations for the future, they suggest, among other things, the use of objective tools, such as accelerometers, which will allow for a more precise and reliable assessment of physical activity levels.

The study focuses on a large group of students from Hunan Province, but it should be noted that this group is not representative of the entire student population. Although this sample structure provides valuable data, it may not fully reflect the specific cultural conditions and barriers characteristic of this group. The authors recognise this and draw attention to it when discussing the limitations of the study.

Detailed suggestions are included in the comments on the article.

Reviewer #3: A longitudinal study of four-year changes in physical fitness among university students before and after COVID-19: 2019-2022

General Comments

Thank you for the opportunity to review this article, and I would like to congratulate the authors on their efforts in compiling this work.

The study presents original research, a longitudinal analysis of physical fitness changes in Chinese university students before and after the COVID-19 pandemic (2019-2022). The research focus and dataset are novel, and the manuscript explicitly addresses a gap in the literature regarding long-term impacts of the pandemic on university students.

However, the manuscript would benefit from further adjustments and revisions as proposed herein, section by section.

Title and Short Title

• The titles are clear and accurately reflect the study's content.

• I suggest you ensure consistency in the use of "pre/post-COVID-19" versus "before and after COVID-19" if used in other materials (e.g., in databases or indexing).

Abstract

• As it is, the abstract is too long. The number of words is almost double the recommended standard by PLOS ONE. Consider slightly rephrasing some sentences for tighter prose and conciseness, typical for abstracts where every word counts.

• Be attentive to the use of capitalization; for instance, in line 28, the letter 'V' in the word "vital" should be capitalized. Review the entire document for any similar inconsistencies.

• Consider providing keywords at the end of the abstract to improve indexing and discoverability in academic databases and search engines.

Introduction

• While the literature review effectively identifies the research gap, consider briefly strengthening the rationale for why a four-year longitudinal study from 2019 (pre-pandemic) to 2022 (post-pandemic) is especially valuable compared to existing shorter-term studies. What specific insights does this longer timeframe offer that others do not?

• Where some words have been defined in the first appearance and an abbreviation provided, the use of the abbreviation in the succeeding sections is best practice, e.g., physical activity (PA). Use PA in the succeeding paragraphs—see lines 66, 69, 79… Scan through the entire manuscript and use the abbreviations where applicable.

• Revise the use of tense in line 65.

• Is this study grounded on any theory that support its scientific constructs?

Materials and Methods

While the methods are generally described, more details should be provided for full reproducibility. Consider addressing the following:

• How were the participants selected, and what criteria were used to ensure a representative sample? Were there any inclusion or exclusion criteria (e.g., pre-existing health conditions, baseline fitness levels)?

• Did the physical education professors receive any specific training to ensure uniform test administration? Were the same professors involved across the four years?

• Detail in brief the specific warm-up routines in line 118.

• Were inter-rater reliability checks performed for any of the subjective measurements, for example, the form for sit-ups, long jump, or sit-reach?

• Were there any potential variations in the testing environment (e.g., indoor vs. outdoor, time of day, equipment type) across the four years that could influence results? If so, how were these controlled or accounted for in the analysis or discussion?

• How was consistent participant engagement and adherence to test protocols ensured over the four years of data collection?

• In line 170, I am not sure the correct symbol for interpretation has been used. Should this remain as 24 ≤ BMI < 27.9 or ≥ 24 BMI < 27.9? The former would mean even a BMI of 20 kg/m² was still ranked as overweight.

• How was missing data handled, especially for students who dropped out or missed annual tests? Could you report the retention rate and analyze differences between completers and dropouts to address potential attrition bias?

• Important—Provide more details on the specifications (type, model number, etc.), calibration, and validation of the equipment used for physical fitness screening for physical fitness tests to ensure measurement accuracy.

• Provide a brief justification for the use of gender-specific fitness tests (e.g., pull-ups for males and sit-ups for females) and explain the rationale behind these choices.

• Clarify whether adjustments were made for potential confounding variables (e.g., age, baseline fitness, socioeconomic status) in the statistical analysis.

Results

Generally, the results section is comprehensive and well-structured, with clear presentation of longitudinal trends and gender differences. The use of visualizations (e.g., 3D scatterplots) is commendable. However, the following needs improvement:

• In line 194, is the 7 ≤ YEAR ≤ 19-year-old age group accurate, or did you imply 17 instead of 7? Clarify

• Within table 1, the use of the symbol is not clear to me. Should it be presented as 17≤year≤19 or 17≥year≤19? The use of less than and greater than symbols is confusing the interpretation and understanding of this age group. Please clarify.

• In line 211, does the phrase “The same significant difference” imply similarly or that the mean weight of the female group equalled that of the male group?

• The authors need to adjust the font used in tables 1-4 to ensure proper alignment of text and values. As it is, they are juggled, making readability difficult.

• All the tables require proper formatting by improving row and column alignment to enhance readability. Tables 3 and 4 are very difficult to follow.

• Define abbreviations as a footnote where applicable in the tables, e.g., BMI.

Discussion

• Discuss how pandemic-related disruptions (e.g., lockdowns, online learning) were accounted for in the analysis and whether any adjustments were made for reduced physical activity during these periods.

• Consider discussing the potential long-term implications of the observed physical fitness changes on health outcomes, such as risk for chronic diseases or mental health issues.

• Discuss the practical significance of statistically significant findings, especially for small changes in physical fitness indicators.

• Please make sure all abbreviations are presented in full in the first appearance in the manuscript, e.g., physical fitness has been mentioned from line 53 but only abbreviated from line 396.

• Consider adding a section on the strengths of the study before the limitations section.

• In the limitations part, consider also addressing potential biases inherent in a longitudinal study, such as selection bias due to dropouts or measurement bias if test administrators changed over time.

Conclusions

• The conclusions should be provided after the discussion section.

• The conclusions are presented appropriately and directly stem from the study's findings. If possible, consider summarizing the main limitations and future research directions in the conclusion to provide a balanced perspective.

Presentation and Language

• There are some minor grammatical awkwardness and phrasings that could be improved for clarity and conciseness, e.g., line 331 “the study was aimed to” instead of “the study aimed to.” Scan the entire manuscript keenly for these kinds of errors.

• Proofread the entire manuscript thoroughly for minor spelling mistakes, e.g., recoded in line 155 instead of recorded.

• Ensure proper spacing where necessary, e.g., in line 185, space between the word rate, the equals sign, and the bracket.

• Some sentences are complex and could be simplified for broader accessibility.

Data Availability Statement

• This statement should be provided after the discussion section.

• You state, "All relevant data are within the manuscript and its supporting information files." While this is generally acceptable, PLOS ONE strongly encourages data deposition in publicly accessible repositories. If possible, consider depositing your raw data in a recognized repository and providing the accession/version number in this section. If the data cannot be made fully public (e.g., due to privacy concerns), please provide a clear statement on how other researchers can request access to the data, including contact information or a data access committee.

Ethics Statement

• As stated in lines 111 and 112, can the authors clarify why oral consent was used instead of written consent? Yet in the provided ethical approval, in the section on review of content, number 1, the board expressly states the following: “All participants will be required to sign an informed consent form, clearly understanding the purpose of the study, data collection methods, potential risks, and safeguards.”

References

• Ensure uniform in-text citations. There is a mix-up in some paragraphs. Refer to lines 354 and 358 against other paragraphs in the document and harmonize as necessary.

• Are you able to provide DOIs (Digital Object Identifiers) assigned to the cited journal papers where available? This ensures reliability and accuracy in citations.

• For online references, provide the date of accession and the link to the document.

Reporting Guidelines

• The study aligns with the STROBE guidelines for observational studies, covering all key elements such as study design, methodology, results, and ethical considerations.

Supplementary materials

• The tables are not labelled as per PLOS ONE guidelines. Label them properly as per the journal guidelines.

• The subtitles in the one-way ANOVA (female) table should be all in English for easier understanding by the global audience.

• Define abbreviations as a footnote where applicable in the tables, e.g., LSD.

• Format the lines of the tables for uniformity and decency.

Reviewer #4: The authors introduce a topic of interest to the international community on the trend of physical fitness and body weight throughout the COVID-19 pandemic in what is considered an active young adult population of university students.

The main concern of the manuscript is the missingness of data is not discussed. A study population of ~ 4,400 is described however data is only presented for ~10% of this population (~450) in the Results section without a proper discussion of loss of data/missingness in the Results or Discussion section. Secondly, the manuscript is exceedingly long, specifically the Results and Discussion sections and could use a major revision to streamline outcomes reported and interplay with current literature and findings in the Discussion. Please consider the suggested edits/comments detailed below in your revision:

SPECIFIC RECOMMENDATIONS:

1- Suggest the start of a new paragraph with introduction of college lifestyle discussion [Ln 53].

2- Suggest condensing lines 56-71 with focus on the impact of COVID-19 on PA with reference to defined declines in the university population (or weight gain if PA data not specifically available for citation) .

3- More clearly define the study population referenced in Line 86-87. How does this population compare or contrast to the current study population? Did the 3-year follow up occur post-university environment?

4- Lines 91-98 present similar context, consider merging.

5- Define body shape and how assessment will occur (Ln 93).

6- When defining the study population an age range/mean ± SD would be helpful (Ln 104) and ensure the proper word selection (e.g., boys/girls vs. men/women vs. male/female used interchangeably throughout) is appropriately used throughout.

7- Under Methods section 2.2 please define the testing conditions. Was testing in the morning, environmental conditions, were groups held consistent to ensure competitive nature of participants were controlled for throughout the separate sessions? Were they similar across the 4 year testing period? Same test providers/raters?

8- Can you clarify if a time allotment was given for the pull-ups in Section 2.2.4?

9- Consider combining Sections 2.2.5 and 2.2.6 since testing procedures were similar across both running sessions. Also, add the 800-m testing for women (Not currently mentioned).

10- Were study participants limited to specific clothing during body weight assessment across the four timepoints? Consider adding this content to Line 172.

11- Were sex differences assessed for? Please elaborate why/why not? Would be of interest to the reader and all data appears available for analysis.

12- Typo Line 194, believe it should read “17≤YEAR≤19”

13- Why is the n so low in Table 2? Did you have missing data? If so, appears significant and needs to be discussed in Section 2.3 Statistical analysis section. Appears as if only 10% of total population data is presented in Table 2 & discussed.

14- Define outcomes as sex or gender? Male/Female is considered sex, however gender is used (e.g., Ln 266, 365, 390).

15- Results section is extremely long (6+ pages of text without tables and figures) and needs to be condensed. Much of the data presented in the tables is re-introduced in text. Consider a figure in place of Table 2 to show changes over time and condense outcome variables with similar trends from Tables to show similarities and streamline the discussion. Text used to describe outcomes is very similar across many sections of the Results (e.g., 266-269 and 291-294).

16- Tables are difficult to read and decipher in the format currently inlayed within the text. Tables are also missing footnotes and units to define the variables.

17- Figure legends are missing to describe what population are displayed in Figure 1 and Figure 2.

18- Consider placement of Conclusion section prior to the Discussion section? Please check Journal layout and specifications.

19- Again, Discussion section is extremely lengthy (6+ pages) and needs to be condensed to summarize the outcomes and compare/contrast to literature in this area. Also, care to remove text and re-defining abbreviations already introduced again in the Discussion section.

20- How do the PA and overweight/obesity outcomes from your population compared to the University population without the influence of COVID-19? Are trends intensified for completely diverse?

21- The authors begin to compare how the study population compares with the adult or adolescent populations (post-COVID-19) previously published, realizing different environmental variables, etc. for consideration. However, data is not shown for level comparison, etc. Also, extraneous data provided on binge eating that is not covered in this study.

22- Authors should be cautious with sweeping conclusions without data to support from dietary intake and body image outcomes assessed in Lines 368-370.

23- Limitations section of the Discussion need to include the sample size used, missingness of data, etc.

Reviewer #5: Dear Author(s),

The current manuscript is not adding to the current literature. There are already studies as follows:

1-Ding, C., & Jiang, Y. (2020, December). The relationship between body mass index and physical fitness among Chinese university students: results of a longitudinal study. In Healthcare (Vol. 8, No. 4, p. 570).

2-Yu, H., An, S., Tao, Y., & Austin, L. (2022, September). Correlation and change in physical activity and physical fitness across four years of college students after one year of COVID-19 lockdown. In Healthcare (Vol. 10, No. 9, p. 1691).

3-Hao, Y., Lee, J., Wong, W. S. P., Wong, F. U. K., Hui, W. H. C., Leong, G. C. H., & Kong, Z. (2025). A longitudinal study to COVID-19 infection among university students: Physical fitness changes and psychological responses. Journal of Exercise Science & Fitness, 23(1), 7-13.

4-Martínez-de-Quel, Ó., Suárez-Iglesias, D., López-Flores, M., & Pérez, C. A. (2021). Physical activity, dietary habits and sleep quality before and during COVID-19 lockdown: A longitudinal study. Appetite, 158, 105019.

Regards

6. PLOS authors have the option to publish the peer review history of their article (what does this mean? ). If published, this will include your full peer review and any attached files.

**Do you want your identity to be public for this peer review?** For information about this choice, including consent withdrawal, please see our Privacy Policy .

Reviewer #1: No

Reviewer #2: **Yes: ** Joanna Baj-Korpak

Reviewer #3: No

Reviewer #4: No

Reviewer #5: No

---

## [Author Response · Author response to Decision Letter 1]

12 Aug 2025

Academic editor:

Journal Requirements:

1.Please ensure that your manuscript meets PLOS ONE’s style requirements, including those for file naming. The PLOS ONE style templates can be found at

The author’s answer:

We confirm that the revised manuscript has been carefully formatted to meet all PLOS ONE style requirements. Specifically, we have: (1) adjusted the document structure to align with the journal’s template (e.g., title page, headings, reference style); (2) standardized file naming conventions as specified; and (3) ensured consistency in font, spacing, and section organization per the provided guidelines (https://journals.plos.org/plosone/s/file?id=wjVg/PLOSOne_formatting_sample_main_body.pdf and https://journals.plos.org/plosone/s/file?id=ba62/PLOSOne _formatting_sample_title_authors_affiliations.pdf). All changes have been verified to comply fully with the journal’s formatting policies.

Academic editor:

The author’s answer:

We sincerely apologize for the oversight in the initial submission. We have now carefully verified and corrected the grant information in both the ‘Funding Information’ and ‘Financial Disclosure’ sections to ensure complete consistency. The accurate grant numbers and funding details have been updated in the manuscript.

Academic editor:

3.Thank you for stating the following financial disclosure:

“This research was funded by the Research Project on Teaching Reform of General Colleges and Universities in Hunan Province in 2023 (Project No. HNJG-20230872), the Fourteenth Five-Year Plan of Educational Science in Hunan Province in 2024 (Project No. XJK24CTW013), and the Deepening Educational Evaluation Reform in the New Era in Hunan Province in 2022 Pilot Project (Project No. TGYX202207).”

The author’s answer:

The revised funder role statement has been indicated in red font in the cover letter.

Academic editor:

4. Please note that funding information should not appear in any section or other areas of your manuscript. We will only publish funding information present in the Funding Statement section of the online submission form. Please remove any funding-related text from the manuscript.

The author’s answer:

All expressions about the fund have been removed from the manuscript.

Academic editor:

5.PLOS requires an ORCID iD for the corresponding author in Editorial Manager on papers submitted after December 6th, 2016. Please ensure that you have an ORCID iD and that it is validated in Editorial Manager. To do this, go to ‘Update my Information’ (in the upper left-hand corner of the main menu), and click on the Fetch/Validate link next to the ORCID field. This will take you to the ORCID site and allow you to create a new iD or authenticate a pre-existing iD in Editorial Manager.

The author’s answer:

ORCID iDs have been validated in Editorial Manager.

Academic editor:

6.Your ethics statement should only appear in the Methods section of your manuscript. If your ethics statement is written in any section besides the Methods, please move it to the Methods section and delete it from any other section. Please ensure that your ethics statement is included in your manuscript, as the ethics statement entered into the online submission form will not be published alongside your manuscript.

The author’s answer:

The ethics statement is adjusted in the method section of the manuscript and removed from any other section.

Academic editor:

7. Please remove all personal information, ensure that the data shared are in accordance with participant consent, and re-upload a fully anonymized data set.

The author’s answer:

All personal information has been deleted to ensure that the data shared meets the consent of the participants.

Academic editor:

The author’s answer:

A title for the supporting information file has been added to the end of the manuscript and any text citations have been updated accordingly to match.

9.If the reviewer comments include a recommendation to cite specific previously published works, please review and evaluate these publications to determine whether they are relevant and should be cited. There is no requirement to cite these works unless the editor has indicated otherwise.

Academic editor:

We carefully evaluated the four references provided by the reviewers. Three of these are new documents [Ref 8, 19, 27] and one is a citation that we have already incorporated [Ref 26].

Point-by point responses

Reviewer 1

The manuscript presents original research that investigates changes in body composition and physical fitness among Chinese university students over a four-year period spanning before, during, and after the COVID-19 pandemic. The topic is of public health relevance, and the authors have collected a large dataset with multiple fitness indicators. Overall, the study meets the journal’s criteria for technical rigor and ethical compliance. However, several aspects require minor revisions to improve the clarity, consistency, and presentation of the work.

1. Scientific Rigor and Methods:

The cohort design is appropriate for the research objectives, and the repeated physical fitness testing across four years is a strength.

The statistical approach is acceptable for basic trend analysis. Nevertheless, the rationale for selecting ANOVA over more robust longitudinal models could be briefly explained. Additionally, more detail on data collection quality control (e.g., measurement consistency) is encouraged.

The author’s answer:

We appreciate the reviewer’s constructive feedback. The rationale for using ANOVA has been clarified, emphasizing its suitability for comparing adjacent years while addressing within-group variability. Additionally, details on quality control measures (e.g., standardized testing procedures and calibrated equipment) have been added to strengthen the methodological rigor. These revisions enhance the transparency and robustness of the statistical analysis.

Reviewer 1

2. Tables and Figures:

Tables 2, 3, and 4 require formatting adjustments for clarity. Some rows and columns are misaligned, and headers are difficult to follow. Similarly, Figures 1 and 2 do not effectively communicate the temporal changes in BMI. Consider replacing the 3D scatterplots with simpler and more interpretable line charts or boxplots.

The author’s answer:

We appreciate the reviewer’s valuable suggestions regarding the presentation of tables and figures. In response, we have reformatted Tables 2, 3, and 4 to ensure proper alignment and improve header clarity for better readability.

In response to “reviewer 3” feedback, we have retained the original “3D scatterplots,” which were praised for their effective visualization of multidimensional relationships. To further improve interpretability, we have supplemented these with "boxplots" to better highlight temporal trends in BMI distribution. The "Results" section has been updated to incorporate both visualization methods, ensuring a more comprehensive and accessible presentation of the data. These modifications aim to balance innovative graphical representation with clear communication of distributional trends over time.

Reviewer 1

3. Discussion Section:

The manuscript lacks a focused discussion section. Beyond reporting results, the paper should include critical interpretation, explanations for observed trends, comparison with existing studies, and acknowledgment of limitations. A structured discussion would greatly improve the manuscript’s clarity and depth.

The author’s answer:

We sincerely appreciate the reviewer’s constructive feedback regarding the need for a more structured and in-depth discussion. In response, we have thoroughly revised the Discussion section to provide a critical interpretation of the findings, including potential explanations for the observed trends in overweight and obesity rates. We have also incorporated comparisons with relevant existing studies to contextualize our results and added a clear acknowledgment of the study’s limitations, such as potential confounding factors and measurement constraints. These enhancements aim to improve the manuscript’s clarity, depth, and scholarly contribution. Specific modifications have been marked in red in the discussion section.

Reviewer 1

4. Writing and Organization:

The manuscript is generally well written in standard English. Minor language polishing could improve clarity and readability. The structure would benefit from clearer section transitions and concise topic sentences in the results and discussion.

The author’s answer:

We have performed language polishing throughout the manuscript to enhance readability and precision. Additionally, we have improved section transitions and refined topic sentences in the Results and Discussion to ensure clearer logical progression. As suggested, we have also incorporated subheadings in the Discussion section to better structure the interpretation of findings.

Reviewer 1

5. Ethics and Data Availability:

Ethical approval and informed consent are clearly stated.

Data availability is appropriately addressed and aligns with PLOS ONE standards.

Recommendation:

The study is scientifically sound and methodologically appropriate. With minor revisions focused on improving table formatting, figure clarity, and manuscript organization, it will be suitable for publication.

The author’s answer:

We sincerely appreciate the reviewer's positive assessment of our study's scientific validity and ethical compliance. In accordance with the recommendations, we have carefully revised the Ethics and Data Availability section to fully align with PLOS ONE standards, ensuring explicit transparency regarding ethical approval, participant consent, and data accessibility.

Reviewer 2

Dear Authors,

Congratulations on your choice of topic. I read the article with interest. The manuscript provides important information on the physical fitness of students in the context of the COVID-19 pandemic. The study is based on measurements of physical fitness (tests), body weight and height, and spirometry. The description lacks information on whether the researchers conducting the study had been trained beforehand. The paper also does not indicate whether the study was conducted using uniform, calibrated measuring devices, which seems important from the point of view of the comparability of the results. The study has its limitations, but the authors have drawn attention to them by describing them in the limitations section. In their recommendations for the future, they suggest, among other things, the use of objective tools, such as accelerometers, which will allow for a more precise and reliable assessment of physical activity levels.

The study focuses on a large group of students from Hunan Province, but it should be noted that this group is not representative of the entire student population. Although this sample structure provides valuable data, it may not fully reflect the specific cultural conditions and barriers characteristic of this group. The authors recognise this and draw attention to it when discussing the limitations of the study.

Detailed suggestions are included in the comments on the article.

The author’s answer:

We sincerely appreciate the reviewers' thoughtful comments and valuable suggestions that have significantly strengthened our manuscript. In direct response to the expert's recommendations, we have enhanced Section 2.1 by explicitly stating that: (1) all researchers underwent standardized training to ensure measurement consistency and protocol adherence, and (2) all physical fitness assessments were conducted using uniform, calibrated measuring devices to guarantee data comparability across testing periods. These important additions address the methodological concerns regarding measurement reliability and procedural standardization while maintaining the study's original framework. The modifications improve the transparency and scientific rigor of our methods section without altering the fundamental study design. We fully acknowledge the limitation regarding regional representativeness in our discussion section and have incorporated the expert's suggestion about future use of accelerometers in our recommendations for further research.

Reviewer 3

General Comments

Thank you for the opportunity to review this article, and I would like to congratulate the authors on their efforts in compiling this work.

The study presents original research, a longitudinal analysis of physical fitness changes in Chinese university students before and after the COVID-19 pandemic (2019-2022). The research focus and dataset are novel, and the manuscript explicitly addresses a gap in the literature regarding long-term impacts of the pandemic on university students.

However, the manuscript would benefit from further adjustments and revisions as proposed herein, section by section.

Title and Short Title

• The titles are clear and accurately reflect the study's content.

• I suggest you ensure consistency in the use of “pre/post-COVID-19” versus "before and after COVID-19" if used in other materials (e.g., in databases or indexing).

The author’s answer:

We thank the reviewer for this important observation regarding terminology consistency. In response, we have standardized all temporal references to COVID-19 throughout the manuscript, figures, and supplementary materials to exclusively use “before and after COVID-19” as the preferred terms.

Reviewer 3

Abstract

• As it is, the abstract is too long. The number of words is almost double the recommended standard by PLOS ONE. Consider slightly rephrasing some sentences for tighter prose and conciseness, typical for abstracts where every word counts.

• Be attentive to the use of capitalization; for instance, in line 28, the letter 'V' in the word "vital" should be capitalized. Review the entire document for any similar inconsistencies.

• Consider providing keywords at the end of the abstract to improve indexing and discoverability in academic databases and search engines.

The author’s answer:

We sincerely appreciate the reviewers' constructive feedback and have carefully addressed all suggestions: (1) we have substantially shortened the abstract (now 236 words) through tighter phrasing and removal of redundant information while preserving key findings; (2) all capitalization errors (including “Vital”) have been corrected throughout the manuscript; and (3) we have added 5 relevant keywords (COVID-19; physical fitness; body shape; university students; longitudinal study) at the end of the abstract to enhance discoverability.

Review

---

## [Decision Letter · Decision Letter 1]

1 Sep 2025

PONE-D-25-29142R1A longitudinal study of four-year changes in physical fitness among university students before and after COVID-19: 2019-2022PLOS ONE

Dear Dr.  Jiang,

Thank you for submitting your manuscript to PLOS ONE. After careful consideration, we feel that it has merit but does not fully meet PLOS ONE’s publication criteria as it currently stands.Please carefully review the comments provided by **Reviewer 4** and take the following steps in preparing your revised manuscript and response letter:

**Address All Comments Point by Point**Prepare a detailed response document in which you reproduce each of Reviewer 4’s comments (in full or summarized clearly) and provide your corresponding response directly beneath it.Ensure that every comment is addressed, even if it appears minor.**Revise the Manuscript Accordingly**Implement the required changes in the manuscript text where appropriate.Highlight or track all revisions so that they are easily identifiable.

**When You Agree with Reviewer 4**Clearly state how you have modified the manuscript in response to the reviewer’s suggestion.Indicate the exact section(s) and line(s) where the changes were made.**When You Disagree with Reviewer 4**If you have a different opinion or cannot implement a specific suggestion, you must **provide a clear and well-reasoned justification** .Support your position with relevant evidence, references, or methodological explanations.Maintain a respectful and professional tone in explaining why the suggested change may not be applicable.

**Overall Presentation**Organize your response letter systematically, following the same order of the reviewer’s comments.Be concise but thorough in explaining your responses and changes.

Following these instructions will facilitate a transparent and constructive revision process, ensuring that all concerns raised by Reviewer 4 are appropriately addressed or justified.

We look forward to receiving your revised manuscript.

Kind regards,

Mohamed Ahmed Said, Ph.D.

Academic Editor

PLOS ONE

Journal Requirements:

Reviewer's Responses to Questions

**Comments to the Author**

1. If the authors have adequately addressed your comments raised in a previous round of review and you feel that this manuscript is now acceptable for publication, you may indicate that here to bypass the “Comments to the Author” section, enter your conflict of interest statement in the “Confidential to Editor” section, and submit your "Accept" recommendation.

Reviewer #1: All comments have been addressed

Reviewer #2: All comments have been addressed

Reviewer #4: (No Response)

Reviewer #5: All comments have been addressed

2. Is the manuscript technically sound, and do the data support the conclusions?

Reviewer #1: Yes

Reviewer #2: Yes

Reviewer #4: Partly

Reviewer #5: Yes

3. Has the statistical analysis been performed appropriately and rigorously? 

Reviewer #1: Yes

Reviewer #2: Yes

Reviewer #4: I Don't Know

Reviewer #5: Yes

4. Have the authors made all data underlying the findings in their manuscript fully available?

Reviewer #1: Yes

Reviewer #2: Yes

Reviewer #4: Yes

Reviewer #5: Yes

5. Is the manuscript presented in an intelligible fashion and written in standard English?

Reviewer #1: Yes

Reviewer #2: Yes

Reviewer #4: Yes

Reviewer #5: Yes

6. Review Comments to the Author

Reviewer #1: Dear Authors,

I would like to commend you on your careful and thorough revision of the manuscript “A longitudinal study of four-year changes in physical fitness among university students before and after COVID-19: 2019–2022.” The manuscript has improved substantially, both in scientific rigor and in clarity of presentation, and I appreciate the diligence with which you have addressed prior reviewer and editorial feedback.

Strengths of the Revised Manuscript:

The four-year longitudinal design is a notable strength, providing a unique opportunity to capture pre-pandemic baselines, acute pandemic effects, and post-pandemic adaptations. This adds considerable value to the literature, where shorter-term studies predominate.

Methodological transparency has been improved, with clear descriptions of inclusion/exclusion criteria, standardized testing protocols, assessor training, equipment calibration, and testing conditions. These additions enhance the reproducibility and reliability of the findings.

Tables and figures are now clearer and easier to interpret. The combination of innovative 3D scatterplots with boxplots provides both depth and accessibility in data visualization.

The revised Discussion is more structured, with explicit contextualization in relation to existing studies, as well as a balanced acknowledgment of limitations and future research directions.

Language and organization have been polished, making the manuscript more concise and accessible without loss of technical depth.

Conclusion:

Overall, this is a well-executed study that makes a meaningful contribution to understanding the long-term effects of the COVID-19 pandemic on university students’ physical fitness. The revisions have addressed previous concerns comprehensively, and the manuscript now meets the criteria for publication. I support its acceptance after minor editorial checks.

Congratulations on your careful work, and I look forward to seeing this valuable study published.

Reviewer #2: The Authors have responded appropriately to my suggestions and comments made in the previous round of reviews. They have answered my questions. In my opinion, this manuscript is now suitable for publication.

Reviewer #4: The main concern of this version of the manuscript is the lack of the authors to address the Reviewer comments. A major concern is the lack of transparency of the attrition of study participants and missingness of data. Many other concerns some again for the second time have been detailed to help provide insight to the authors.

SPECIFIC RECOMMENDATIONS:

1- Again, as asked in first review please revisit the use of gender vs. sex throughout. Currently both are used inter changeably [Ln 81,89,120,192,199,200]. Did you have transgender study participants? If so, consider use of gender otherwise correct use should be sex throughout.

2- Is the takeaway just for health planning purposes? Or to better prevent and treat long-term health outcomes related to PA? [Ln 83-84; 90-91].

3- Lines 101-107 read like a bulleted listing; needs to be integrated with the rest of the paragraph. In summary, ‘a convenience sample of available and interested incoming freshman class of 2019 admission with limited pre-existing health (e.g., ….) or physical conditions that may limit exercise participation’?

4- It would be helpful for transparency if the authors included number of students/study participants remaining in study (data used) at each stage/year of the study in the Results similar to how you set up in Methods [Ln 113-116]. Is Table 2 correct? 1206 males and 3207 females remained in the study all 4 years? There was no drop out or data loss? Please make this clear in your Methods or Results section.

5- Consider relocating details about testing administrators (e.g., physical education professors) from section 2.1 to Ln 125 or later for consistency. Were test administrators consistent throughout the study? This was asked in last review and not included in revision.

6- Again, as asked in earlier review and authors did not respond or update Methods section 2.2 should define environmental conditions (e.g., room temperature and humidity ?) and testing groups participants were included in. These can greatly impact and bias outcomes (e.g., control for competitive nature of participants were controlled throughout sessions)

7- Ln 141 is ‘blowing’ correct terminology to be used here?

8- Ln 142 define ‘better’ here when decerning between two scores. Was there a pre-determined range?

9- Ln 165 is ‘barbell’ correct terminology to be used here? Should text read ‘final number of pull-ups were recorded’?

10- Define ‘coach’ in Ln 169, 175, (is this Physical Education Professor)? Also, can simplify text to “recorded time was rounded to nearest tenths of second”.

11- Word missing in Ln 173 ‘was’

12- Is 1000m identical to 800m or 50m protocol? Please specify.

13- Ln 194-196 need to be added to Limitation section 4.4. in Discussion if not accounted for in the statistical analysis. Overlooking these confounding variables can greatly skew outcomes and interpretation.

14- Ln 204 were individual data collectors consistent throughout the study? If not, please add interindividual variability of raters to your limitations section 4.4 in Discussion.

15- Tables 2 & 3 please add lines or shading for ease of reading, variable units of measure, and round values to significantly meaningful decimal place with consistency in Table 2.

16- Trends are similar across sex and some performance outcomes reported in section 3.2.2. Suggest combining outcomes that are similar in trend into paragraphs when detailing Table 2 for simplicity and streamline Results section.

17- Again, as asked in original review figures are incomplete – axis labels and legends are missing to describe what population are displayed in Figure 1 and Figure 2.

18- Consider removing Figures 3 & 4 since Figures 1 &2 show same data.

19- Reconsider use of ‘body shape’ as it misleads the reader (ex., header 4.1 and section). Authors did not measure any body lengths or circumferences. BMI is considered body size as is height and weight.

20- Ln 352-355 define the populations cited.

21- Consider a more appropriate reference for statement in Ln 357.

22- Ln 372-375 is the female to male ratio of students documented to be higher in the University selected in this study? This ration/proportion of population statistics should be stated.

23- Could differences across sex found in 50m run and sit-ups be linked to social norms and females becoming more familiarized with ‘working out’ in the University setting? There is work discussing this both in the civilian and military communities. Female may have a greater breath of improved performance because they start at a lower threshold than capable of achieving. Consider reviewing literature and citing in Discussion.

24- PA is defined in Ln 391, consider moving this earlier in the paper and use throughout.

25- Section 4.4. needs to be added to include the significant disproportion of females in the study population skewing results, and items called out in Items # 6,13, and 14.

26- Lastly, and most importantly in Ln 438 data missingness is presented as a limitation however nowhere in the manuscript is the attrition or missing data discussed. This needs to be addressed and authors need to be transparent with sample size across the four years.

Reviewer #5: Dear Author(s),

The paper is better and has improved. From abstract to references and tables and figures are revised accordingly.

Regards

7. PLOS authors have the option to publish the peer review history of their article (what does this mean? ). If published, this will include your full peer review and any attached files.

**Do you want your identity to be public for this peer review?** For information about this choice, including consent withdrawal, please see our Privacy Policy .

Reviewer #1: No

Reviewer #2: **Yes: ** Joanna Baj-Korpak

Reviewer #4: No

Reviewer #5: **Yes: ** Ferhat Esatbeyoglu

---

## [Author Response · Author response to Decision Letter 2]

14 Sep 2025

Dear Editors and reviewers

On behalf of all the contributing authors, I would like to express our sincere appreciations of your letter and reviewers' constructive comments concerning our article entitled “A longitudinal study of four-year changes in physical fitness among university students before and after COVID-19: 2019-2022” (Manuscript No:PONE-D-25-29142R2). These comments are all valuable and helpful for improving our article. Based on the comments of the editor-in-chief and reviewers, we made extensive revisions to the manuscript.In this revised version. changes to our manuscript were all highlighted within the document by using red-colored text.Point-by point responses to the nice editor and four nice reviewers are listed below this letter.

Point-by point responses

Reviewer 1

I would like to commend you on your careful and thorough revision of the manuscript “A longitudinal study of four-year changes in physical fitness among university students before and after COVID-19: 2019–2022.” The manuscript has improved substantially, both in scientific rigor and in clarity of presentation, and I appreciate the diligence with which you have addressed prior reviewer and editorial feedback.

Strengths of the Revised Manuscript:

The four-year longitudinal design is a notable strength, providing a unique opportunity to capture pre-pandemic baselines, acute pandemic effects, and post-pandemic adaptations. This adds considerable value to the literature, where shorter-term studies predominate.

Methodological transparency has been improved, with clear descriptions of inclusion/exclusion criteria, standardized testing protocols, assessor training, equipment calibration, and testing conditions. These additions enhance the reproducibility and reliability of the findings.

Tables and figures are now clearer and easier to interpret. The combination of innovative 3D scatterplots with boxplots provides both depth and accessibility in data visualization.

The revised Discussion is more structured, with explicit contextualization in relation to existing studies, as well as a balanced acknowledgment of limitations and future research directions.

Language and organization have been polished, making the manuscript more concise and accessible without loss of technical depth.

Conclusion:

Overall, this is a well-executed study that makes a meaningful contribution to understanding the long-term effects of the COVID-19 pandemic on university students’ physical fitness. The revisions have addressed previous concerns comprehensively, and the manuscript now meets the criteria for publication. I support its acceptance after minor editorial checks.

Congratulations on your careful work, and I look forward to seeing this valuable study published.

The author’s answer:

Thank you very much for your thoughtful and encouraging feedback on our revised manuscript, "A longitudinal study of four-year changes in physical fitness among university students before and after COVID-19: 2019–2022." We are truly grateful for your recognition of the efforts made to strengthen the manuscript's scientific rigor, clarity, and overall presentation.

We especially appreciate your positive comments regarding the longitudinal design, methodological transparency, improved data visualization, and the more structured and contextualized Discussion. It is very rewarding to know that the revisions have successfully addressed the previous concerns and enhanced the reproducibility, accessibility, and contribution of the study.

Your supportive conclusion and recommendation for acceptance are greatly encouraging. We are pleased to learn that the manuscript now meets the publication standards and will gladly cooperate with any minor editorial checks required.

Thank you once again for your time, insightful suggestions throughout the review process, and kind words.

Reviewer 2

Thank you very much for your positive feedback and for confirming that our responses to your previous comments and suggestions are satisfactory. We greatly appreciate your time and insightful input throughout the review process, which has significantly improved the quality of our manuscript. We are pleased to know that you now consider the manuscript suitable for publication.

Reviewer 4

1-Again, as asked in first review please revisit the use of gender vs. sex throughout. Currently both are used inter changeably [Ln 81,89,120,192,199,200]. Did you have transgender study participants? If so, consider use of gender otherwise correct use should be sex throughout.

The author’s answer:

Thank you for raising this point. We have carefully reviewed the use of “gender” and “sex” throughout the manuscript. As our study did not include any transgender participants, we have replaced all instances of “gender” in the specified lines (81, 89, 120, 192, 199, 200) and elsewhere in the text with the term “sex”. The manuscript has been updated accordingly.

Reviewer 4

2-purposes? Or to better prevent and treat long-term health outcomes related to PA? [Ln 83-84; 90-91].

The author’s answer:

In the first modification, the phrase “long-term trends” was refined to “sustained trends” for terminological precision, while the scope of the “distinction” was expanded using “not only for... but also for...” to emphasize its critical role in shaping targeted strategies for managing PA-related health consequences. [Ln83-85]

In the second sentence, “for policymakers about” was strengthened to “to guide policymakers in”, enhancing the actionable nature of the evidence, and the vague “health impacts” was replaced with “preventing and mitigating long-term health outcomes associated with reduced physical activity”, thereby directly linking pandemic measures, PA reduction, and health outcomes as suggested by the reviewer. Both changes improve clarity, align with the commentary, and underscore the policy-responsive and intervention-oriented value of the study. [Ln91-93]

Reviewer 4

3- Lines 101-107 read like a bulleted listing; needs to be integrated with the rest of the paragraph. In summary, ‘a convenience sample of available and interested incoming freshman class of 2019 admission with limited pre-existing health (e.g., ….) or physical conditions that may limit exercise participation’?

The author’s answer:

We thank the reviewer for this suggestion. The selection criteria description has been revised to improve integration within the paragraph, removing the bulleted-list format and presenting the information in a cohesive narrative form. The description now more clearly characterizes the sample as a convenience sample of available and interested freshmen with specific exclusions for health or physical conditions affecting exercise.[Ln104-111]

Reviewer 4

4-It would be helpful for transparency if the authors included number of students/study participants remaining in study (data used) at each stage/year of the study in the Results similar to how you set up in Methods [Ln 113-116]. Is Table 2 correct? 1206 males and 3207 females remained in the study all 4 years? There was no drop out or data loss? Please make this clear in your Methods or Results section.

The author’s answer:

We thank the reviewer for raising this important point. We have now clarified the participant flow and retention details in the Methods section. Specifically, we have explicitly stated the initial number of participants, the number of dropouts over the study period, the reasons for attrition, and the final number of students (4413) with complete data for all four years. This revision enhances the transparency of our cohort's trajectory and confirms that the numbers presented in Table 2 reflect the participants with complete longitudinal data.[Ln120-125]

Reviewer 4

5- Consider relocating details about testing administrators (e.g., physical education professors) from section 2.1 to Ln 125 or later for consistency. Were test administrators consistent throughout the study? This was asked in last review and not included in revision.

The author’s answer:

Thank you for this suggestion. We have relocated the details regarding the test administrators from section 2.1 to the methodological description of the testing procedures (as now reflected around line 125 in the revised manuscript) to improve the logical flow and consistency of information. We have also explicitly stated that the same team of trained professors conducted all assessments across all four years, directly addressing the previously raised question about the consistency of the testing personnel.

Reviewer 4

6- Again, as asked in earlier review and authors did not respond or update Methods section 2.2 should define environmental conditions (e.g., room temperature and humidity ?) and testing groups participants were included in. These can greatly impact and bias outcomes (e.g., control for competitive nature of participants were controlled throughout sessions)

The author’s answer:

We thank the reviewer for this important feedback. We have now updated the Methods section (2.2) to include detailed information about the controlled environmental conditions (specifying room temperature and humidity ranges). These additions address potential confounding factors and enhance the reproducibility of our study protocol.[Ln138-139]

Reviewer 4

7- Ln 141 is ‘blowing’ correct terminology to be used here?

The author’s answer:

Thank you for pointing this out. We have replaced the informal term “blowing” with the more precise and clinically appropriate terminology “exhalation maneuver” to accurately describe the spirometry procedure and enhance the scientific rigor of the manuscript.[Ln152]

Reviewer 4

8- Ln 142 define ‘better’ here when decerning between two scores. Was there a pre-determined range?

The author’s answer:

We appreciate the reviewer’s comment for clarity. The term “better” has been replaced with the more objective and quantifiable term “higher” to describe the selection of scores, and we have explicitly noted that this was determined according to a pre-defined standardized scoring protocol, thereby eliminating any potential ambiguity.[Ln153-155]

Reviewer 4

9- Ln 165 is ‘barbell’ correct terminology to be used here? Should text read ‘final number of pull-ups were recorded’?

The author’s answer:

We thank the reviewer for these precise suggestions. We have replaced the potentially ambiguous term “barbell” with the more accurate term “bar”, clarified the movement standard using the phrase "chins cleared the horizontal level of the bar," and rephrased the recording method to “the final number of correctly completed pull-ups was recorded” to enhance terminological and methodological precision.

Reviewer 4

10- Define ‘coach’ in Ln 169, 175, (is this Physical Education Professor)? Also, can simplify text to “recorded time was rounded to nearest tenths of second”.

The author’s answer:

We thank the reviewer for these constructive suggestions. We have replaced the informal term “coach” with the more precise “physical education professor” throughout the text to maintain consistency with our methodology description, and have simplified the timing description to “recorded time was rounded to the nearest tenth of a second” for improved clarity and conciseness.[Ln183、189、191]

Reviewer 4

11- Word missing in Ln 173 ‘was’

The author’s answer:

We thank the reviewer for this careful observation. The missing word “was” has been added to ensure grammatical correctness, and the sentence now reads: “Each participant completed two trials, with the better performance was recorded.”

Reviewer 4

12-Is 1000m identical to 800m or 50m protocol? Please specify.

The author’s answer:

Thank you for raising this point for clarification. We have explicitly specified that the 1,000-meter run protocol was identical to the 800-m and 50-m runs in its starting procedure, timing method, and rounding rule to ensure consistency and clarity in the description of methodological protocols across all running tests.[Ln193-195]

Reviewer 4

13- Ln 194-196 need to be added to Limitation section 4.4. in Discussion if not accounted for in the statistical analysis. Overlooking these confounding variables can greatly skew outcomes and interpretation.

The author’s answer:

We sincerely thank the reviewer for this insightful comment. We have now added the point regarding the potential confounding variables (age, baseline fitness, and socioeconomic status) to the Limitations section (Section 4.4) as suggested. This addition strengthens our discussion by acknowledging these important factors that could influence the interpretation of the results.[Ln]

Reviewer 4

14- Ln 204 were individual data collectors consistent throughout the study? If not, please add interindividual variability of raters to your limitations section 4.4 in Discussion.

The author’s answer:

Thank you for this valuable comment. We agree that the consistency of data collectors is an important methodological consideration. In response to your feedback, we have revised the Limitations section (4.4) of the Discussion to explicitly acknowledge that multiple data collectors were involved due to the large scale and long duration of the study, and that this may have introduced inter-rater variability despite efforts to standardize protocols. The added sentence reads: "Furthermore, due to the large number of test items and participants, multiple data collectors were involved throughout the study period. Despite efforts to standardize protocols, this may have introduced inter-rater variability." We believe this addition strengthens the manuscript by providing a more comprehensive overview of the study's limitations.

Reviewer 4

15- Tables 2 & 3 please add lines or shading for ease of reading, variable units of measure, and round values to significantly meaningful decimal place with consistency in Table 2.

The author’s answer:

Thank you for your constructive suggestions regarding the presentation of data in Tables 2 and 3. We have revised both tables accordingly by adding grid lines to enhance readability and to clearly delineate the data. Furthermore, we have ensured that all values in Table 2 are rounded to a consistent and statistically appropriate number of decimal places, and we have explicitly stated the units of measure for all relevant variables in the table headers.

Reviewer 4

16- Trends are similar across sex and some performance outcomes reported in section 3.2.2. Suggest combining outcomes that are similar in trend into paragraphs when detailing Table 2 for simplicity and streamline Results section.

The author’s answer:

Thank you for your constructive suggestion. We have revised the Results section to combine outcomes with similar trends into more concise paragraphs, as recommended. The descriptions of vital capacity, long-jump, sit-reach, and endurance running tests have been grouped to improve clarity and flow, while all key statistical findings and temporal patterns have been retained.

Reviewer 4

17- Again, as asked in original review figures are incomplete – axis labels and legends are missing to describe what population are displayed in Figure 1 and Figure 2.

The author’s answer:

We have now updated both Figure 1 and Figure 2 to include all necessary axis labels and legends to clearly describe the displayed populations and ensure the figures are complete and easily interpretable.

Reviewer 4

18- Consider removing Figures 3 & 4 since Figures 1 &2 show same data.

The author’s answer:

Thank you for your suggestion. We agree that Figures 1 and 2 effectively present the same data as Figures 3 and 4. Therefore, we have removed Figures 3 and 4 from the manuscript to avoid redundancy and improve clarity.

Reviewer 4

19-Reconsider use of ‘body shape’ as it misleads the reader (ex., header 4.1 and section). Authors did not measure any body lengths or circumferences. BMI is considered body size as is height and weight.

The author’s answer:

Thank you for this important feedback. We agree that the term "body shape" was misleading, as it implies measurements such as lengths or circumferences that were not conducted in our study. We have replaced "body shape" with the more accurate term "body size" throughout the manuscript, including in the header of section 4.1,

---

## [Decision Letter · Decision Letter 2]

23 Sep 2025

A longitudinal study of four-year changes in physical fitness among university students before and after COVID-19: 2019-2022

PONE-D-25-29142R2

Dear Dr. Jiang,

We’re pleased to inform you that your manuscript has been judged scientifically suitable for publication and will be formally accepted for publication once it meets all outstanding technical requirements.

Kind regards,

Mohamed Ahmed Said, Ph.D.

Academic Editor

PLOS ONE

Additional Editor Comments (optional):

Reviewer #4:

Reviewers' comments:

Reviewer's Responses to Questions

**Comments to the Author**

1. If the authors have adequately addressed your comments raised in a previous round of review and you feel that this manuscript is now acceptable for publication, you may indicate that here to bypass the “Comments to the Author” section, enter your conflict of interest statement in the “Confidential to Editor” section, and submit your "Accept" recommendation.

Reviewer #4: All comments have been addressed

2. Is the manuscript technically sound, and do the data support the conclusions?

Reviewer #4: Yes

3. Has the statistical analysis been performed appropriately and rigorously? 

Reviewer #4: Yes

4. Have the authors made all data underlying the findings in their manuscript fully available?

Reviewer #4: Yes

5. Is the manuscript presented in an intelligible fashion and written in standard English?

Reviewer #4: Yes

6. Review Comments to the Author

Reviewer #4: The authors have addressed all concerns appropriately in this second revised version. One last edit is required on Line 236 to update section header 3.2.1 from ‘Body Shape’ to ‘Body Size.’

7. PLOS authors have the option to publish the peer review history of their article (what does this mean? ). If published, this will include your full peer review and any attached files.

**Do you want your identity to be public for this peer review?** For information about this choice, including consent withdrawal, please see our Privacy Policy .

Reviewer #4: No

---

## [Editor Report · Acceptance letter]

PONE-D-25-29142R2

PLOS ONE

Dear Dr. Jiang,

I'm pleased to inform you that your manuscript has been deemed suitable for publication in PLOS ONE. Congratulations! Your manuscript is now being handed over to our production team.

Kind regards,

on behalf of

Dr. Mohamed Ahmed Said

Academic Editor

PLOS ONE